# Strong External Electric Fields Reduce Explosive Sensitivity: A Theoretical Investigation into the Reaction Selectivity in NH_2_NO_2_∙∙∙NH_3_

**DOI:** 10.3390/molecules28062586

**Published:** 2023-03-13

**Authors:** Fu-De Ren, Ying-Zhe Liu, Xiao-Lei Wang, Li-Li Qiu, Zi-Hui Meng, Xiang Cheng, Yong-Xiang Li

**Affiliations:** 1School of Chemical Engineering and Technology, North University of China, Taiyuan 030051, China; 2State Key Laboratory of Fluorine and Nitrogen Chemicals, Xi’an Modern Chemistry Research Institute, Xi’an 710065, China; 3School of Chemistry and Chemical Engineering, Beijing Institute of Technology, Beijing 100081, China; 4School of Intelligent Engineering, Zhengzhou University of Aeronautics, Zhengzhou 450003, China

**Keywords:** external electric field, reaction selectivity, explosive sensitivity, intermolecular hydrogen exchange concerted reaction, surface electrostatic potential

## Abstract

Controlling the selectivity of a detonation initiation reaction of explosive is essential to reduce sensitivity, and it seems impossible to reduce it by strengthening the external electric field. To verify this, the effects of external electric fields on the initiation reactions in NH_2_NO_2_∙∙∙NH_3_, a model system of the nitroamine explosive with alkaline additive, were investigated at the MP2/6-311++G(2d,p) and CCSD(T)/6-311++G(2d,p) levels. The concerted effect in the intermolecular hydrogen exchange is characterized by an index of the imaginary vibrations. Due to the weakened concerted effects by the electric field along the −*x*-direction opposite to the “reaction axis”, the dominant reaction changes from the intermolecular hydrogen exchange to 1,3-intramolecular hydrogen transference with the increase in the field strengths. Furthermore, the stronger the field strengths, the higher the barrier heights become, indicating the lower sensitivities. Therefore, by increasing the field strength and adjusting the orientation between the field and “reaction axis”, not only can the reaction selectivity be controlled, but the sensitivity can also be reduced, in particular under a super-strong field. Thus, a traditional concept, in which the explosive is dangerous under the super-strong external electric field, is theoretically broken. Compared to the neutral medium, a low sensitivity of the explosive with alkaline can be achieved under the stronger field. Employing atoms in molecules, reduced density gradient, and surface electrostatic potentials, the origin of the reaction selectivity and sensitivity change is revealed. This work provides a new idea for the technical improvement regarding adding the external electric field into the explosive system.

## 1. Introduction

Explosive sensitivity is an index by which to measure the stability of explosives when exposed to external stimuli. Today, it has been imperative to explore effective ways to reduce the sensitivity of the explosive under the external electric field. This is mainly initiated and driven by two factors: (1) technical improvement of the microelectric explosion and (2) avoidance of accidental explosion from electrostatic spark. The microelectric explosion is a new explosion technology in which the external electric field is added to the ignition and initiation system to improve performance, such as detonation heat, detonation velocity, detonation pressure, etc., and it has aroused great interest [1,2,3,4]. However, the sensitivities of the explosives loaded in the wire or slapper plate explosive detonators are often increased under the external electric fields, leading to serious obstacles in technical improvements to adapt to the modern bad war environment. Moreover, static electricity cannot be avoided, and for explosives the potential catastrophic accidental explosion from electrostatic spark is possible anywhere and anytime. It is generally believed that the stronger the external electric field, the more easily the detonation initiation occurs, leading to a higher explosive sensitivity and a higher risk of accidental explosion [5,6,7,8]. Under a strong external electric field, it seems impossible to achieve low sensitivity with a high-energetic explosive or even a sensitivity slightly lower than that without an electric field. 

Fortunately, however, the external electric field has been widely used for over two decades as a “smart reagent” to control reactivity by adjusting the orientation of the electric field to the “reaction axis” [9,10,11,12,13,14,15,16,17,18,19,20,21,22,23,24,25]. “Reaction axis” is defined as a direction along which a chemical bond is formed or broken. When the direction of the external electric field is the same as that of the “reaction axis”, the field will promote the electronic reorganization beneficial to the reaction resulting from reactants bonding to products, leading to a decreased barrier height and an acceleration of the reaction. On the contrary, when the direction of the external electric field is opposite, the field will inhibit the formation of the bonding mode from the reactant to product, leading to an increased barrier height [26,27]. Therefore, by adjusting the orientation of the external electric field to “reaction axis”, the barrier heights and reaction paths can be changed, and thus the regioselectivity and reaction rate can be controlled [14,15,16,17,18,19,20,21,22,23,24,25,26,27]. Many theoretical [19] and experimental investigations [20,28] have been carried out to control reactivity by adjustment of the orientation of the external electric fields, extending from the general catalytic reactions [23,29] to the selectivity of enzymatic-like bond activations [12,22], DNA damage [11], proton transfer [9], photochemical CO_2_ capture [25], etc. In two recent perspective articles [13,26], Shaik et al. demonstrated that the external electric field could control redox or nonredox reactions, and the corresponding barrier height could be increased by enforcing the electron transfer from acceptor to donor along/against the external-electric-field direction. Furthermore, it is well known that there are often multiple reaction axes, and the changes of reactivities along the different reaction axes are often asynchronous with the changes of the strength and direction of the external electric field [30]. Some are sensitive, and some are insensitive to the external electric fields. This gave us the inspiration that, by adjusting the strength and orientation of the external electric field along the “reaction axis” of explosive molecules, such as those involving the homolysis of the C−NO_2_, N−NO_2_, and O−NO_2_ trigger linkages; the hydrogen transference; etc., as the first step in the initiation reaction [31,32], the reaction path could be changed, and the barrier height could be increased, leading to decreased explosive sensitivity. 

In fact, reducing the explosive sensitivity by adjusting the external electric field to control the selectivity of the detonation initiation reaction in experiment [33] and theory [4,5,6] has aroused great interest. An electromagnetic pulse effect was tested during the bridge wire electric explosion [2]. We have found that, for the simple (pure) explosives, the bond dissociation energies (BDEs) of the “trigger linkages” and barrier heights of the initiation reactions could be increased under certain external electric fields, and the selectivity of the detonation initiation reaction could be controlled, and thus the explosive sensitivities were reduced by the adjustment of the strength or orientation of the external electric fields [34,35,36]. For the mixture of nitroamine explosive with H_2_O, NH_2_NO_2_∙∙∙H_2_O as a model system, an abrupt jump of the activation energy between the intermolecular and 1,3-intramolecular hydrogen transfer reactions was confirmed. Therefore, we predicted that, by controlling the strength or orientation of the external electric field, the explosive sensitivity could be reduced in a neutral environment, such as aqueous solution [30]. 

Alkaline agents and additives with the −NH_2_ group have more influence on decomposition rates and sensitivities of explosives than neutral or acidic species [37,38]. Using a special Bourdon manometer, Shu et al. investigated the thermal decomposition of a series of mixtures with a concentration of RDX (cyclotrimethylene trinitramine) in the range of 0.1~2%. They found that the decomposition rate of RDX was hardly affected in a neutral medium, such as benzene, isooctane, naphthalene, etc., and compared with molten RDX, the change in rate was less than 5.0% [37]. On the other hand, it increased 2.8~21.0 times in the alkaline medium, such as the agent with the −NH_2_ or −NHR group [39]. We found that, in the alkaline environment, the impact sensitivities of explosives were even increased or decreased by more than 20% due to either the modified BDEs of the “trigger bonds” or the changed activation energies or hydrogen-transfer reaction paths induced by the intermolecular interactions between the explosives and additives [40,41]. Therefore, the mechanism of the sensitivity change is more unpredictable for the explosive in the alkaline environment, and it is more imperative to reveal the essence of the explosive sensitivity and reduce it by adjusting the external electric field for the explosive in the alkaline environment than that in the neutral or acidic medium. 

The nitramine explosive is one of the most widely used energetic materials. In this work, NH_2_NO_2_ and NH_3_ are selected as the model compounds of the nitroamine explosive and alkaline additive to reveal the essence of the explosive sensitivity under the external electric fields and explore ways to reduce the explosive sensitivity in the alkaline environment. The fragments of the initial decomposition reaction of NH_2_NO_2_ were determined as O•, NH_2_NO•, NH_2_N•, •NH_2_, •NO_2_, NO_2_, and NO [42], and the competitive reaction between the N−NO_2_ bond cleavage and NH_2_NO_2_ → NH_2_ONO rearrangement was also confirmed [43]. For NH_2_NO_2_∙∙∙NH_3_ under the external electric field, the first or rate-determining step of the detonation reaction may be the homolysis of the N−NO_2_ bond, rearrangement, 1,3-intramolecular hydrogen transfer, or intermolecular hydrogen exchange between NH_2_NO_2_ and NH_3_. Therefore, the effects of the external electric fields on the intermolecular hydrogen exchange and 1,3-intramolecular hydrogen transfer as well as homolysis of the N−NO_2_ bond in NH_2_NO_2_∙∙∙NH_3_ will be mainly investigated with theoretical methods, accompanied by a comparison with those of the NH_2_NO_2_∙∙∙H_2_O system. One of our goals is to clarify whether or not, by strengthening external electric field, the sensitivity of the high-energetic nitroamine explosive in the alkaline environment can be reduced to lower than that without an electric field. This work is useful for microelectric explosion technology to rationally design equipment, to efficiently add the external electric fields into the nitramine explosive systems with the alkaline agents or additives, and to avoid catastrophic explosions in the process of preparation, transportation, and use under external electric fields. 

## 2. Results and Discussion

Two conformations of NH_2_NO_2_∙∙∙NH_3_ were optimized. In one conformation, NH_3_ is not only a H-bonded donor, but also as a H-bonded acceptor (or Lewis base), shown by the intermolecular N−H∙∙∙O_2_N and N−H∙∙∙NH_3_ H-bonds (see Figure 1). In another conformation, NH_3_ is only a H-bonded donor with one intermolecular N−H∙∙∙O_2_N H-bond (see Appendix A). The electron energy of the former is 32.0 kJ/mol less than that of the latter at the MP2/6-311++G(2d,p) level, so the former was chosen as the reactant. 

Two feasible initial reactions were predicted: intermolecular hydrogen exchange with TS1 and 1,3-intramolecular hydrogen transference with TS2. The hydrogen exchange is a concerted reaction, in which one of the H atoms of NH_3_ is transferred to –NO_2_, and simultaneously one of the H atoms of –NH_2_ moiety in NH_2_N(O)OH• is transferred to •NH_2_ radical, i.e.,
NH_2_NO_2_∙∙∙NH_3_ ↔ NH_2_N(O)OH•∙∙∙•NH_2_ (biradical) ↔ NHN(O)OH∙∙∙NH_3_

For 1,3-intramolecular hydrogen transference, the H atom of the –NH_2_ moiety is transferred to −NO_2_ to form NHN(O)OH∙∙∙NH_3_. In no field, the barrier of TS1 is far lower than that of TS2 or BDE of the N–NO_2_ bond (see Appendix A), as the “trigger linkage” of the nitroamine explosive [44]. The intermolecular hydrogen exchange occurs preferentially, similar to the detonation initiation mechanism of the CH_3_NO_2_∙∙∙H_2_O [45] and NH_2_NO_2_∙∙∙H_2_O systems [30]. The following will give a comparison of the effect of the external electric field on the kinetics of the intermolecular hydrogen exchange with 1,3-intramolecular hydrogen transference and N−NO_2_ bond cleavage. Although the NH_2_NO_2_→NH_2_ONO rearrangement was confirmed [43], the barrier is too high (about 380.0 kJ/mol at the MP2/6-311++G(2d,p) level), so it is not considered in this work. 

### 2.1. Cooperativity of H-Bonds in Reactant under External Electric Field

The fields parallel to the *z*- and *x*-axis directions affect the structures of NH_2_NO_2_∙∙∙NH_3_ considerably more than those parallel to the *y*-axis. Since NH_2_NO_2_ and NH_3_ are both the electron donor and acceptor, and both the O8∙∙∙H7 and N6∙∙∙H5 H-bonds are in the *xz*-plane, the effects of the fields along the *z*- and *x*-axes on the electron transfers corresponding to the O8∙∙∙H7 H-bond are opposite to N6∙∙∙H5. Thus, the field effects on their distances are also opposite. For example, the fields along the +*z*-direction lengthen the O8∙∙∙H7 distance while it is shortened in the −*z*-direction. Therefore, when one H-bonding interaction is strengthened, the other will be weakened, and vice versa. The phenomenon in which multiple intermolecular interactions enhance each other’s strength when they work simultaneously in a system is termed as the cooperativity effect [46]. Due to the effects of the external electric fields, the cooperativity effect of two H-bonding interactions is weakened, leading to an unobvious change of the total intermolecular interactions. 

### 2.2. Concerted Effect of Intermolecular Hydrogen Exchange in External Electric Field

(1)Structures of TS1

The intermolecular hydrogen exchange is mainly accompanied by the changes of the activation O8∙∙∙H7, H7∙∙∙N6, N6∙∙∙H5, and H5∙∙∙N1 distances involving TS1 (see Figure 1). Since the *y*-direction is approximately perpendicular to the plane of the motion regions of the activation atoms H7 and H5, the activation distances are more affected by the external electric fields parallel to the *x*- and *z*-axis directions than by those parallel to the *y*-axis, and the effects of the fields parallel to the *x*-axis are the most notable (see Appendix A). A good (*R*^2^ > 0.9900) linear correlation is found between the change of the N6∙∙∙H5 distances and field strengths along the *z*-axis (see Appendix A). 

On the whole, with the increased electric field along the +*x*-direction, the H5∙∙∙N1 and O8∙∙∙H7 distances are increased while the N6∙∙∙H5 and H7∙∙∙N6 distances are decreased, and the reverse trend occurs along the −*x*-directions. Some structural changes inconsistent with the overall trend have attracted interest. Along the +*x*-direction, when the field strength is larger than +0.004 a.u., the O8∙∙∙H7 and H7∙∙∙N6 distances are hardly changed, indicating that the effect of the external electric field is “immune” to the reaction involving the activation H7 atom. Along the −*x*-direction, several transience values are confirmed in the activation distances. For example, when the field strength is up to −0.010 a.u., the O8∙∙∙H7 and H5∙∙∙N1 distances are increased suddenly while the N6∙∙∙H5 distances are decreased, a local “bump” appearing in the overall trend. Another transience of the O8∙∙∙H7 distance is found (increased suddenly) under the field with the strength of −0.019 a.u. When the field strength is larger than −0.014 a.u., the change trends of the H5∙∙∙N1 and N6∙∙∙H5 distances are the same as those with field strength lower than −0.008 a.u. These results indicate that the change of the activation distances is transient, and the trend is zigzag under the external electric fields with the field strengths larger than −0.010 a.u. This may lead to diversification of the chemical reaction path under the “strong” external electric fields, i.e., either H5 or H7 transfer plays a key role in the main reaction, which, to our knowledge, has not been clearly emphasized in the experimental literature. 

The field effects on the activation distances are confirmed with the AIM results (see Figure 2). The changes of the electron densities *ρ* of the bond critical point (BCP) (3, −1) corresponding to O8∙∙∙H7, H7∙∙∙N6, N6∙∙∙H5, and H5∙∙∙N1 in the fields parallel to the *y*-axis are not significant in comparison with those in the fields along the *x*- and *z*-axes, among which the change corresponding to O8∙∙∙H7 is the most notable under the field along the −*x*-axis. The changes of the *ρ* values within the bond paths related to the activation distances are more notable than those corresponding to other bond paths. The transient change of the interatomic activation distances and the zigzag trend are also confirmed by the AIM results. For example, along the −*x*-direction, an extremely sharp abrupt change of *ρ*_(O8∙∙∙H7)_ from 0.2792 a.u. with the field strength of −0.018 a.u. to 0.1632 a.u. with the strength of −0.019 a.u. is then increased to 0.3028 a.u. with the strength of −0.020 a.u.

(2)Barrier, imaginary vibration, and rate constant of hydrogen exchange

Both the energies of TS1 and reactant decrease synchronously (become more negative) with the increase of the external electric field strength in *z*-directions, so the changes of the activation energies are not obvious (see Figure 3 and Appendix A). Although the energies of TS1 decrease and those of the reactant increase along the +*x*- and *y*-direction, their changes are slight, so the decreased barrier heights are also not obvious. With the field strength of 0.010 or −0.010 a.u., the change of the barrier compared with that in no field is no more than 25.0 kJ/mol in the field along the *z*-, +*x*-, or *y*-axis directions at two levels of theory. However, since the energies of TS1 increase greatly while those of the reactant decrease, significant increased barrier heights are found along the −*x*-direction. With the field strengths from 0.000 to −0.020 a.u., the barrier heights soar from 68.6 to 291.3 kJ/mol at the MP2/6-311++G(2d,p) level. The change is up to more than 100.0 or 220.0 kJ/mol with the field strength of −0.010 or −0.020 a.u., accompanied by a relative value of more than 140.0% or 300.0%. In general, the change of the barrier compared with that in no field is no more than 15.0 kJ/mol with the field strength of −0.010 or +0.010 a.u. [34,35,36,44].

It is well known that the unrestricted single reference methods give rise to spin contamination when applied to open-shell systems since the unrestricted Hartree–Fock wave function is not an eigenfunction of the total spin operator, i.e., the expectation value ˂*S*^2^> may not be equal to S(S+1), leading to an inaccurate energy. For the restricted open-shell Hartree–Fock calculations with the right ˂*S*^2^>, the unphysical results are often generated due to artificially ruined spin polarization. Therefore, it is necessary to use multireference methods for highly spin-contaminated systems with an inherently multireference nature, especially for the transition state [47,48]. For the open-shell systems calculated using the MP2 and CCSD methods, although the treatment of the electron correlation should lower spin contamination, the spin contamination also occurs at the correlated level [49]. Furthermore, although the CCSD method significantly outperforms MP2 in describing systems with a strongly spin-contaminated reference because the cluster single excitation operator partly accounts for the orbital relaxation effects [50,51,52], CCSD(T), they are often rather sensitive to the spin contamination of the reference [53,54]. In this work, for the single-point energy calculation for the reactants and transition states for the hydrogen exchanges in TS1 and TS2 at the UMP2/6-311++G(2d,p) and UCCSD(T)/6-311++G(2d,p)//MP2/6-311++G(2d,p) levels, we obtained <*S*^2^> ≈ 0.0 (vs. expected S(S+1) = 0) in the different external electric field strengths and orientations, including those in the absence of field. Therefore, the impact of spin contamination and multireference character on the initiation reactions in NH_2_NO_2_∙∙∙NH_3_ in the external electric field, as well as in the absence of field, can be ignored. 

The changes of the barriers induced through the external electric fields are also reflected in the imaginary vibrations, Gibbs energies, and rate constants. The field parallels to the *x*-direction have the more notable effect on the magnitude than those along the *z*- or *y*-direction (see Figure 3 and Table 1 and Appendix A). In most cases, the imaginary vibration and rate constant are hardly changed along the *z*- or *y*-direction. For example, the changes of the imaginary vibration are not more than 50 cm^−1^ with the strongest field strength. However, the imaginary vibrations are decreased or increased significantly in the fields along the −*x*-direction, in particular in the fields with strengths more than −0.014 a.u. In comparison with the value in no field, the change is up to 755.9 and 1270.1 cm^−1^ with the field strength of −0.014 and −0.020 a.u., respectively. The rate constants at 298.15 K (*k*_298.15 K_) are decreased remarkably with the increased field strength, from 2.86 × 10^0^ s^−1^ in no field to 4.41 × 10^−20^ s^−1^ with the field strength of −0.010 a.u. and 1.15 × 10^−36^ s^−1^ with −0.020 a.u. Similar to *k*_298.15 K_, for the rate constants at 688.0 K (*k*_688.0 K_), an obvious decrease is also found, reduced 1.87 × 10^21^ times from the field strength of 0.000 to −0.020 a.u. Except for the field strength more than −0.014 a.u., the Wigner corrections are small, and the tunneling-corrected effects on the rate constants could be ignored. 

(3)Concerted reaction and reaction axis of hydrogen exchange

The concerted reaction refers to the reaction in which there is only one transition state involving all the coexistent multiple reactions enhanced by each other, and in which the breaking of chemical bonds and the formation of new bonds occur simultaneously. For a concerted reaction, each of the coexistent multiple reactions corresponds to one “reaction axis”. For a concerted reaction under the external electric field, the effect of the field on each of the coexisting reactions is different: for the reaction in which the direction of the “reaction axis” is consistent with that of the external electric field, the external electric field will accelerate it, while the external field will inhibit the reaction in which the direction of “reaction axis” is opposite to that of the external electric field. The comprehensive effect of the external field on all the coexistent reactions will be reflected in the change of the barrier height of the concerted reaction. 

The intermolecular hydrogen exchange is in essence a concerted (cooperative) process of two hydrogen transfers (i.e., NH_2_NO_2_∙∙∙NH_3_↔NH_2_N(O)OH•∙∙∙•NH_2_ and NH_2_N(O)OH•∙∙∙•NH_2_↔NHN(O)OH∙∙∙NH_3_), and it can be shown clearly from the changes of the imaginary vibrations under the external electric field. In order to evaluate the vibration intensity of the H7 or H5 atom, the average displacement of H in the imaginary vibration is defined as follows: $$\bar{A_{H}}=\sqrt{\frac{{\left(\Delta X\right)}^{2}+{\left(\Delta Y\right)}^{2}+{\left(\Delta Z\right)}^{2}}{3}}$$
where $\bar{A_{H}}$ means the average displacement of H, and ΔX, ΔY, and ΔZ are the maximum value of the H atom vibration along the *x*-, *y*-, and *z*-axis directions, respectively. Along the –*x*-direction field, both the $\bar{A_{H5}}$ and $\bar{A_{H7}}$ values simultaneously increase with the field strengths from 0.000 to −0.010 a.u., decrease from −0.012 to −0.016 a.u., and again increase from −0.018 to −0.020 a.u., showing a synchronous zigzag trend (see Appendix A). Thus, the concerted effect between the NH_2_NO_2_∙∙∙NH_3_ ↔ NH_2_N(O)OH•∙∙∙•NH_2_ and NH_2_N(O)OH•∙∙∙•NH_2_ ↔ NHN(O)OH∙∙∙NH_3_ reactions is confirmed from the synchronous changes of $\bar{A_{H5}}$ and $\bar{A_{H7}}$. 

Above, two hydrogen transfer processes in the concerted reaction correspond to two “reaction axes” along N6→H7→O8 and N1→H5→N6 (see Figure 1). For the reaction axis along N6→H7→O8, the smaller the electron density of N6, and the higher the electron density of O8, the more likely the activation H7 atom is transferred from N6 to O8. On the contrary, H7 is easily transferred toward N6. When the direction of the external electric field is the same as that of the N6→H7→O8 reaction axis, the electron density of N6 will be decreased, and that of O8 will be increased. As a result, the electric field will induce the electronic reorganization, which is favorable for the transformation from NH_2_NO_2_∙∙∙NH_3_ to NH_2_N(O)OH•∙∙∙•NH_2_. Conversely, the external field hinders this transformation. Similar to N6→H7→O8, for the reaction axis along N1→H5→N6, the smaller the electron density of N1, and the higher the electron density of N6, the more likely H5 is transferred from N1 to N6. The external electric field, which follows the direction of N1→H5→N6, can induce the transformation from NH_2_N(O)OH•∙∙∙•NH_2_ to NHN(O)OH∙∙∙NH_3_. The electric field in the opposite direction leads to the inhibition of the formation of NHN(O)OH∙∙∙NH_3_. 

Under the external electric field, the change of the atomic charge is often complicated due to the influence of the molecular dipole (internal electric field). From the APT charges collected in Appendix A, along the −*x*-direction with the field strengths no more than −0.008 a.u., both of the negative charges of N6 and O8 increase with the increase of the field strengths. When the field strengths are more than −0.008 a.u., the negative charges of O8 decrease while those of N6 decrease first (−0.008 a.u. ~ −0.012 a.u.) and then increase, leading to the more notable negative charge of N6 than that of O8 in the field strength of −0.020 a.u. Thus, the H7 atom with the positive charge would rather bind to N6 than O8 with the increase of the external electric field strength in the −*x*-direction. In other words, the −*x*-direction of the external electric field is opposite to that of the “reaction axis” along N6→H7→O8, and it is not beneficial to the hydrogen transfer from NH_2_NO_2_∙∙∙NH_3_ to NH_2_N(O)OH•∙∙∙•NH_2_ with the increase of the electric field strength. Except for the field strengths of −0.008 a.u. ~−0.012 a.u., the negative charges of N1 are always larger than those of N6, and thus the H7 atom with the positive charge would rather bind to N1 than N6. Therefore, although the −*x*-direction is the same as that of the N1→H5→N6 reaction axis, it is not beneficial to the intermolecular hydrogen transfer of NH_2_N(O)OH•∙∙∙•NH_2_ → NHN(O)OH∙∙∙NH_3_, i.e., this electric field is unfavorable to the NH_2_NO_2_∙∙∙NH_3_→NHN(O)OH∙∙∙NH_3_ concerted reaction. It should be noted that, as mentioned above, in the absence of an electric field, there is a concerted reaction between the N6→H7→O8 and N1→H5→N6 reactions, which promotes the intermolecular hydrogen exchange reaction of NH_2_NO_2_∙∙∙NH_3_→NHN(O)OH∙∙∙NH_3_. Thus, under the electric field along the −*x*-direction, the concerted effect is weakened, leading to the increased barrier heights of the hydrogen exchange reaction. Furthermore, the stronger the external electric field along the −*x*-direction, the more seriously the concerted effect is weakened and the higher the barrier heights become, which is in agreement with the barrier height results shown in Figure 3. 

The weakened concerted effect could also be seen from the changes of the activation distances induced by the external electric fields. Along the *z*- and *x*-axes, the external electric field effect on the O8∙∙∙H7 and H7∙∙∙N6 distances are just opposite each other, as is also found for the N6∙∙∙H5 and H5∙∙∙N1 distances. For example, with the increased electric fields along the +*z*-direction, the O8∙∙∙H7 and H5∙∙∙N1 distances are increased, while the N6∙∙∙H5 and H7∙∙∙N6 distances are decreased. In particular, although the significant effects on the activation distances are achieved along the −*x*-direction, such as transient and zigzag features, the transilience of the barrier is not found. This can be explained as follows: the change of the barrier in the hydrogen exchange reaction is a result of the concerted effect of the two reactions. Although the molecular and electronic structures of the species corresponding to one of the reactions change greatly in a certain electric field, their changes in the other reactions are not obvious, even when they have opposite effects on each other’s barriers, which leads to the weakening of the concerted effect and acts as a buffer effect for the fluctuation of the barriers in the total hydrogen exchange reaction. 

### 2.3. 1,3-Intramolecular Hydrogen Transfer in External Electric Field

The structures of TS2 in 1,3-intramolecular hydrogen transfer are shown in Figure 1. Similar to the intermolecular hydrogen exchange, the fields parallel to the *z*- and *x*-axes affect the activation distances considerably more than those parallel to the *y*-axis. Along the −*z*-direction, the changes of the activation N1∙∙∙H4 and H4∙∙∙O3 distances are irregular. With the increased electric field along the +*z*-direction, the N1∙∙∙H4 distance is increased while the H4∙∙∙O3 distance is decreased. Along the +*x*-direction, the N1∙∙∙H4 and H4∙∙∙O3 distances are increased, and the opposite trend is found along the −*x*-direction. Different from the intermolecular hydrogen exchange, the abrupt change of the activation N1∙∙∙H4 or H4∙∙∙O3 distance is not found under the strong field, and they are increased or decreased gently. Moreover, the changes of the activation distances are far smaller than those of the intermolecular O8∙∙∙H7 and N6∙∙∙H5 distances. For example, the O8∙∙∙H7 distance is up to more than 3.200 Å under the field along the +*z*-axis, indicating that the O8∙∙∙H7 interaction disappears. The changes of the activation distances are confirmed by the RDG results (see Figure 4). A good (*R*^2^ > 0.9850) linear correlation is found between the changes of the activation H4∙∙∙O3 distances and field strengths E_x_ (see Appendix A). These structural changes are confirmed by the electron densities of the bond critical points (BCP) (3, −1). 

The fields parallel to the *z*- and *x*-axes affect the energies considerably more than those parallel to the *y*-axis. Along the +*z*-axis, although the energies of TS2 and reactants decrease with the increase of the electric field, the changes of the former are more significant than those of the latter, leading to decreased barrier heights (see Appendix A). Along the −*z*-axis, due to the decreased energies of the reactant while energies of TS2 increased with the increase of the electric field, the barrier heights increase tremendously, up to more than 230.0 kJ/mol with the field strength of more than −0.010 a.u. Since the energies of TS2 and reactant synchronously increase along the +*x*-axis direction, the change of the barrier height is not obvious. Along the −*x*-axis, although the energies of TS2 and reactant decrease, the change of the latter is far more significant than that of the former, leading to the significant increase of the barriers. 

There is one “reaction axis” along N1→H4→O3 in the 1,3-intramolecular hydrogen transfer from NH_2_NO_2_∙∙∙NH_3_ to NHN(O)OH∙∙∙NH_3_. The smaller the electron density of N1, and the higher the electron density of O3, the more likely that the activation H4 atom will be transferred from N1 to O3, and the smaller the barrier height will be. Conversely, the hydrogen transfer to O3 becomes difficult. From the APT charges in Appendix A, with the increase of field strength along the −*x*-axis, although the negative charges of N1 and O3 increase, the negative charges of N1 are always greater than that of O3, which leads to H4 preferring to bond to N1. Thus, the N1→H4→O3 reaction is inhibited, leading to the increased barriers. 

The fields parallel to the *z*- and *x*-directions have a more notable effect on the magnitudes of the imaginary vibration, Gibbs energies, and rate constant of the 1,3-intramolecular hydrogen transfer than those along the *y*-direction (see Table 1 and Appendix A). The imaginary vibrations are increased under the electric fields along the −*z*- and +*x*-directions; the opposite trend is found in the +*z*- and −*x*-directions; and the decreased values are notable along −*x*-direction, up to 48.6 and 111.2 cm^−1^ with the field strength of −0.010 and −0.020 a.u., respectively. A low imaginary vibration could lead to a small curvature at the region near the TS on the potential energy surface (PES) [55], and thus the fields along the −*z*- and +*x*-directions flatten the PES near TS2. Different from TS1, the Wigner corrections are up to 4.829 at 298.15 K or 1.719 at 688.0 K, suggesting that the tunneling-corrected effects on the rate constants are notable at 298.15 K. The rate constant *k*_298.15 K_ is reduced by 1.19 × 10^13^ times from the field strength of 0.000 to −0.020 a.u., and the change is smaller than that of the intermolecular hydrogen exchange. 

### 2.4. Prediction of Explosive Sensitivity under External Electric Field 

Judging from the initiation reaction of explosives at the molecular level, the sensitivity in no field mainly depends on the barrier of the intermolecular hydrogen exchange. Due to the concerted effect between the NH_2_NO_2_∙∙∙NH_3_→NH_2_N(O)OH•∙∙∙•NH_2_ and NH_2_N(O)OH•∙∙∙•NH_2_→NHN(O)OH∙∙∙NH_3_ reactions, the barrier is very low, suggesting a very high explosive sensitivity. This is in agreement with experimental results [37]. 

Along the *y*-, +*x*-, and +*z*-directions of the electric fields, the variation trends of the barrier heights of the intermolecular hydrogen exchange and 1,3-intramolecular hydrogen transferences are synchronous, and the barriers of the former are always far lower than those of the latter. Along the −*z*-direction, although their trends are opposite, the barrier heights of hydrogen exchange are also far lower than those of 1,3-hydrogen transference. Furthermore, with the field strengths of −0.010 ~ +0.010 a.u., the barrier heights of TS1 are also far lower than the BDEs of the N−NO_2_ bond. Therefore, the hydrogen exchange is always dominant with the field strengths of −0.010 ~ +0.010 a.u. along the *y*-, +*x*- and *z*-directions, and the initiation reaction is the intermolecular hydrogen exchange, which controls the explosive sensitivity. The barriers of the hydrogen exchange are insensitive to the external electric fields along the *y*-, +*x*-, and *z*-directions, i.e., the barriers change slowly (see Appendix A). Therefore, the explosive sensitivities of NH_2_NO_2_∙∙∙H_3_O are almost unchanged, close to those in no field. In other words, by adjusting the field strengths and orientations between the “reaction axes” and external electric fields along the *y*-, *+x*-, or *z*-directions, the selectivity of the reactions could not be changed or controlled, and the sensitivities could not be obviously reduced. 

The conversion of the dominant reactions occurs along the −*x*-direction (see Figure 3, Table 1, Appendix A). The intersection point of the two curves, the barriers of the TS1 and TS2 versus field strengths, is found with the field strengths between −0.008 and −0.010 a.u., and between −0.012 and −0.014 a.u. at the CCSD(T)/6-311++G(2d,p) level. At the MP2/6-311++G(2d,p) level, it is also found between −0.012 and −0.014 a.u. Although both barrier heights of TS1 and TS2 increase with the enhanced field strength along the −*x*-direction, due to the weakened concerted effect in the hydrogen exchange by the strong fields, the change of the barrier heights of TS1 is more notable than that of TS2. When the field strength is lower than −0.008 a.u., the barrier heights of the TS1 are lower than those of TS2. The corresponding rate constants of TS1 are more than 3.89×10^4^ times those of TS2 at the MP2/6-311++G(2d,p) level, and the intermolecular hydrogen exchange is dominant. Due to the significant increase of the barriers of the hydrogen exchange, the sensitivities are obviously reduced. When the field strength is more than −0.014 a.u., the barrier heights of TS1 are higher than those of TS2. The rate constants of TS2 are more than 1.21×10^8^ times those of TS1, and the 1,3-intramolecular hydrogen transference is dominant. Due to the increase of the barriers of the 1,3-intramolecular hydrogen transference, the sensitivities are also reduced. Although the dominant reaction cannot be identified explicitly within the field strengths of −0.008 a.u.~−0.014 a.u., the barriers of the hydrogen exchange and 1,3-intramolecular hydrogen transference are both increased, leading to reduced sensitivities, as well. Therefore, by adjusting the field strengths and orientations between the “reaction axes” and external electric fields along the −*x*-direction, not only can the selectivity of the reaction involving the explosive system NH_2_NO_2_∙∙∙NH_3_ be controlled, but the explosive sensitivities can also be reduced. Thus, it can be inferred that, by adjusting the external field orientation and strengthening the electric field, the sensitivity of the high-energetic nitroamine explosive in the alkaline environment can be reduced greatly to much lower than that without electric field. 

In particular, along the −*x*-direction, the stronger the field strengths, the higher the barrier heights of the dominant reaction, suggesting that the the explosive sensitivities are significantly reduced. For example, compared with the barrier height in no field, a significant increased barrier height is found under the super-strong external electric field, such as −0.020 a.u., suggesting significantly decreased explosive sensitivities under the super-strong external electric field. This is one of the most coveted expectations in the field of explosives: significant reduction of explosive sensitivity induced by a super-strong external electric field. It is well known that the introduction of an external electric field into energetic material can increase the energy and thus accelerate the detonation velocity and increase the detonation pressure [33,56,57]. Then, if the stability can simultaneously also be increased, the inherent contradiction between performance and stability of the explosive will be solved, and not only will a traditional concept, in which the explosives are dangerous under the super-strong external electric field, be broken theoretically, but by strengthening the external electric field, the sensitivity of the high-energetic explosive can also be reduced or even reduced to much lower than that without an electric field. This will be of great significance to the improvement of the technology so that external electric fields are added safely to the energetic material system to enhance explosive performance.

Note that the imposed external field strengths cannot exceed the dielectric strengths of the explosive system. Otherwise, the explosive device may be broken down, and the work to control the selectivity of the reaction and reduce the explosive sensitivity by adjusting the external electric fields will not be practical. Furthermore, although the sensitivity can theoretically be reduced by adjusting the external field orientation and strengths, in practice, the technology may be very difficult because the direction of molecular motion is disorderly, and the control of the direction of the external electric field to the “reaction axis” is one of the technical bottlenecks. There seems to be no way to align an electric field effectively to keep the mixture stable but rather only a portion of the randomly aligned molecules, the rest of which could have their stability reduced and make the device explode. However, the external electric field effect itself of changing the reaction pathways and explosive sensitivity should be remarkable enough to be submitted as a subject of experimental scrutiny in the energetic material field. After all, the orientation of molecules can be achieved by means of Langmuir-Blodgett techniques [58,59], and there has been experimental work in which the external electric field is imposed on energetic material [57]. Furthermore, the external electric field can drive proton transfer [9], change the reaction pathways between different tautomers [10], and control the selectivity of bond activations [12]. Electron transport studies in molecular-scale systems have become possible [60,61], not only in theory [62], but also in experiments [63]. 

### 2.5. Surface Electrostatic Potentials of TS1 under the Field along −x-Orientation

The ESP shows characteristics of electron density distribution by means of the local parameters and global statistical quantities, and it can be used to evaluate the possibility of chemical reaction [64,65]. For example, for an electron-rich atom, the more notable local negative surface ESPs (*V*_s_^−^), the more significant the electron sufficiency within the localized region prone to the electrophilic reaction becomes. According to global parameters, the larger the variance $\sigma_{+}^{2}$ or $\sigma_{-}^{2}$, the greater possibility of the nucleophilic or electrophilic reaction becomes [66,67,68,69]. In order to further reveal the essence of the sharp increase of the barriers of TS1 under the field along −*x*-orientation, the surface electrostatic potentials involving TS1 are shown in Figure 5 and Table 2.

Because the negative ESP of N6 in the NH_3_ moiety is shielded tightly by four positive ESPs of the H atoms, the changes in the chemical bonding modes of the activation H7 to N6 and O8 are explored through the changes of the negative ESP of O8 ($V_{S,O8}^{-}$) under the electric field. With the increase of the electric field strength along the −*x*-orientation, the value of $V_{S,O8}^{-}$ obviously decreases, indicating that the possibility of the formation of the chemical bond between H7 and O8 is reduced greatly. Thus, the concerted effect is weakened in the hydrogen exchange, and the barrier is increased dramatically along the −*x*-direction, as is consistent with the barrier result and charge analysis. 

From the overall trend, with the increase of electric field strength, the absolute values of $\bar{V_{S}^{+}}$, $\bar{V_{S}^{-}}$, $\sigma_{+}^{2}$, and $\sigma_{-}^{2}$ are decreased. However, within the field strengths between −0.008 and −0.012 a.u., except for $\sigma_{-}^{2}$, the other values are all increased suddenly and then decreased, as is also found in the change of the polar surface area (|ESP| > 10 kcal/mol). 

The changes of the global statistical quantities of ESPs, such as $\bar{V_{S}^{+}}$, $\bar{V_{S}^{-}}$, $\sigma_{+}^{2}$ and polar surface area, are more consistent with the those of the activation O8∙∙∙H7, H7∙∙∙N6, N6∙∙∙H5, and H5∙∙∙N1 distances induced by the external electric fields, while the changes of the local parameter $V_{S,O8}^{-}$ and global statistical quantity $\sigma_{-}^{2}$ are more consistent with the trend of the barriers of the concerted reaction. This suggests that the essence of the barrier changes in the concerted reaction is not only originated from the changes of the local electronic properties induced by the external electric fields, but is also related to the changes of the global electronic structures. As mentioned above, a concerted reaction means a reaction in which there is only one transition state involving all the coexistent multiple reactions, and the barrier changes of each of the coexistent reactions must be closely related to the changes of the local electronic structures under the external electric fields. In one word, the coupling effect of the changes of the local and global ESPs, induced by the external electric fields, controls the possibility of the concerted reaction and thus the explosive sensitivity. 

For NH_2_NO_2_∙∙∙H_2_O, with the increase of the field strength along the −*x*-orientation, the negative surface ESPs of the O6 atom of the neutral H_2_O were decreased slightly, while they increased suddenly with the strength of −0.012 a.u. and then decreased gradually. Simultaneously, the positive surface ESPs of the H7 atom were increased slightly, and they increased dramatically with the field strength of −0.013 a.u. and then increased slowly. The electrophilicity of O6 and the nucleophilicity of H7 suddenly were increased dramatically with the strength of −0.012 a.u. or −0.013 a.u., leading to a suddenly and extremely increased barrier of the reaction in which the intermolecular hydrogen transferred from the O6 atom of H_2_O to the O8 atom of the −NO_2_ group. However, for the NH_3_ in NH_2_NO_2_∙∙∙NH_3_, as mentioned above, the negative surface ESP of the N6 atom is shielded by the positive surface ESPs of four H atoms. The change of the electrophilicity of N6 in NH_3_ is less notable than that of O6 in H_2_O induced by the external electric field. Therefore, the barrier of NH_2_NO_2_∙∙∙NH_3_→NHN(O)OH∙∙∙NH_3_ is increased more gently than that of NH_2_NO_2_∙∙∙NH_2_→NHN(O)OH∙∙∙NH_2_, and the sensitivity of the nitramine explosive in the alkaline environment is decreased more gently than that in the neutral medium. In other words, for the nitramine explosive with the neutral medium, low sensitivity can be achieved with a weak external electric field, while it can only be achieved under the super-strong field for the explosive in the alkaline environment. 

## 3. Computational Details

All calculations were carried out with Gaussian 09 programs [70]. The research scheme is shown in Appendix A. Recently, we have used the second order Møller–Plesset perturbation theory (MP2) [71] and coupled cluster theory with single, double, and perturbative triple substitutions CCSD(T) [72] methods with the 6-311++G(2d,p) basis set to investigate the effects of the external electric fields on the hydrogen transference kinetics of the model molecules of energetic material [30,35,36], and we found that the bond dissociation energies (BDEs) of the “trigger linkages” [7] and barrier heights of the initiation reactions associated with hydrogen atom transfer as well as explosive sensitivities [73,74] could be accurately calculated and estimated in the absence and presence of the different external electric fields. Two methods were also used to quantify the electric field effect on the intermolecular interactions [75,76]. Therefore, in this work, the molecular geometries of the reactant and transition state (TS) were fully optimized using the MP2/6-311++G(2d,p) method in the absence and presence of the external electric fields. The energy minima were judged by the criteria of lacking imaginary frequency or having only one imaginary frequency in which two atoms vibrate along the direction of forming or breaking chemical bonds. The activation energies (*E*_a_) were calculated at the MP2/6-311++G(2d,p) and CCSD(T)/6-311++G(2d,p)//MP2/6-311++G(2d,p) levels, respectively. 

CCSD(T) is a high-level electron correlation method [77,78,79], within the chemical accuracy of 1.0 kcal/mol in the total energies [80]. It can be thought of as the “gold standard” in computational chemistry for single-reference systems, in particular in kinetics [81]. Nowadays, the CCSD(T) method has served as a reference in almost any computational approach if affordable, combined with sufficiently large and diffuse basis sets. For example, for hydrogen bonded systems, the accuracy of CCSD(T) is rather high in the dissociation energy, and the error is much lower than for atomization energies, with a RMS error of 0.17 kJ/mol and a relative RMS error of 0.6%, which is a composite method akin to W4 theory. MP2 yields RMS errors of around 1.3 kJ/mol and 6.5%, almost 10 times those of CCSD(T) [82]. Since the external electric field acts on the dipole moment, for MP2 and CCSD(T), the influence of the external electric field on the accuracy is equivalent to the first-order perturbation of energy. 

For the coordinate systems, the N atom of the −NH_2_ group is at the origin, and the N atom of the −NO_2_ moiety is along the +*z*-axis (see Figure 1); the *x*-axis is in the plane composed of the N atom in the −NH_2_ group and two O atoms in the −NO_2_ group, and the N atom of NH_3_ is approximately along the −*x*-axis; and the *y*-axis is defined as the direction perpendicular to the *xz*-plane. In three orthogonal directions, the field strengths of ±0.002, ±0.004, ±0.006, ±0.008, and ±0.010 a.u. were considered. In order to find out a possible conversion of the reaction paths between the intermolecular hydrogen-exchange and 1,3-intramolecular hydrogen-transference reactions, the dynamics were also considered with field strengths of −0.012, −0.014, −0.016, −0.018, −0.019, and −0.020 a.u. along the −*x*-direction.

At the MP2/6-311++G(2d,p) level, the rate constants *k*(*T*) and Wigner tunneling-corrected rate constants *k*_C_(*T*) [83] were estimated at 298.15 and 688.0 K using the conventional transition state theory [84,85], as is expressed with: $$k(T)=\kappa (T)\frac{k_{B}T}{h}\exp(-\frac{\Delta G^{\neq }}{RT})$$
where $k_{B}$, κ(T), *T*, *h*, $\Delta G^{\neq }$, and *R* are the Boltzmann constant, Wigner tunneling correction factor, absolute temperature, Planck’s constant, Gibbs energy change of activation, and universal gas constant, respectively. κ(T) is given as follows [86]: κ(T)=1+124hIm(ν≠)kBT2
where Im($\nu^{\neq }$) is the imaginary frequency corresponding to TS. 

The analyses of the AIM (atoms in molecules) [87] reduced density gradient (RDG) [88] and surface electrostatic potentials (ESPs) [66] were carried out by the Multiwfn programs [89] at the MP2/6-311++G(2d,p) level.

## 4. Conclusions

In order to clarify whether or not, by strengthening the external electric field, the sensitivity of a nitroamine explosive in an alkaline environment can be reduced, the effects of the external electric fields on the initiation reactions in NH_2_NO_2_∙∙∙NH_3_ were investigated using a theoretical method. 

The cooperativity effect of the intermolecular H-bonding interactions of reactant is weakened under the external electric fields. 

The activation distances and barriers of TS1 and TS2 are more affected by the external electric fields parallel to the *x*- and *z*-axes than by those parallel to the *y*-axis, and the effects along the *x*-axis are the most notable. The electric field along the −*x*-direction is unfavorable to the NH_2_NO_2_∙∙∙NH_3_→NHN(O)OH∙∙∙NH_3_ reaction. 

The intermolecular hydrogen exchange is in essence a concerted process, and it can be shown from the change of the imaginary vibrations under the external electric field. The −*x*-direction of the electric field is opposite to that of the “reaction axis” of the dominant reaction in the hydrogen exchange, leading to an inhibition of the reaction from NH_2_NO_2_∙∙∙NH_3_ to NH_2_N(O)OH•∙∙∙•NH_2_. Thus, the concerted effect of the hydrogen exchange is obviously weakened, and the barrier height is increased sharply. It is the weakening or even breaking of the concerted effect in the hydrogen exchange that makes the barrier heights increase dramatically along the −*x*-direction. 

The intermolecular hydrogen exchange is always dominant with the field strengths of −0.010~+0.010 a.u. along the *y*-, +*x*-, and *z*-directions, and it controls the explosive sensitivity. Due to the unobvious barrier changes, the explosive sensitivities are almost unchanged and remain at a high state, close to those in no field. 

However, the conversion of the dominant reaction occurs along the −*x*-direction. The hydrogen exchange and 1,3-intramolecular hydrogen transference are dominant with the field strength lower than −0.008 a.u. and more than −0.014 a.u., respectively. The barriers of both reactions are increased significantly with the increase of the field strengths along the −*x*-direction. Furthermore, the stronger the field strengths, the higher the barrier heights become, suggesting the explosive sensitivities are more significantly reduced. Therefore, by increasing the field strength and adjusting the orientation between the “reaction axes” and external electric fields, not only can the selectivity of the reaction be controlled, but the explosive sensitivities can also be reduced significantly, in particular under the super-strong external electric field. Thus, a traditional concept, in which the explosive is dangerous under a super-strong external electric field, is broken theoretically. 

For a nitramine explosive with a neutral medium, low sensitivity can be achieved with a weak external electric field, and for an explosive in an alkaline environment, low sensitivity can only be achieved under a super-strong field. 

It should be noted that, although the sensitivity can be reduced by theoretically adjusting the external field orientation and strengths, in practice the technology is very difficult because the direction of molecular motion is disorderly, and control of the direction of the external electric field to the “reaction axis” is one of the technical bottlenecks. There seems to be no way to align an electric field effectively to keep the mixture stable but rather only a portion of the randomly aligned molecules, the rest of which could have their stability reduced and have the device explode. However, the external electric field effect itself of changing the reaction pathways and explosive sensitivity should be remarkable enough to be submitted as a subject in the energetic material field. After all, the orientation of molecules can be achieved, and the external electric field can drive the proton transfer and control the selectivity of bond activations. 

This work will be of great significance regarding the improvement of the technology so that external electric fields are added safely into energetic material systems to enhance explosive performance. 

## Figures and Tables

**Figure 1 molecules-28-02586-f001:**
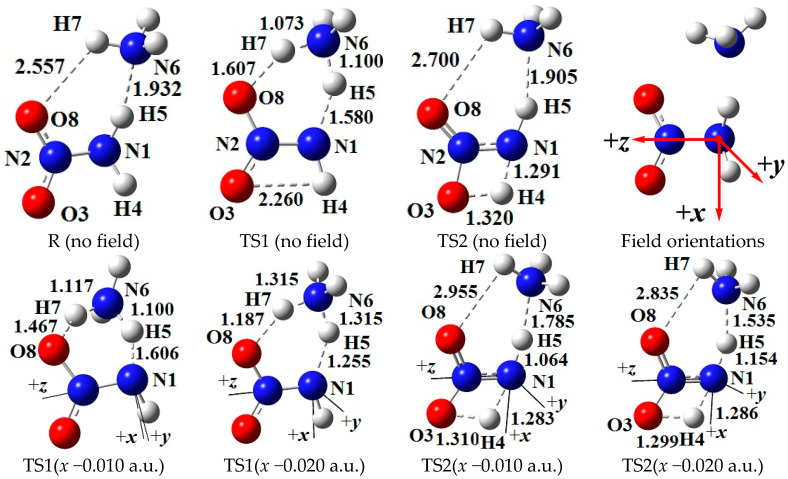
Selected structures of the reactant (R), transition states (TS) in the different external electric field strengths and orientations (including those in the absence of field) at the MP2/6-311++G(2d,p) level (geometric parameters are in Å). The “reaction axes” are N6→H7→O8, N1→H5→N6, and N1→H4→O3.

**Figure 2 molecules-28-02586-f002:**
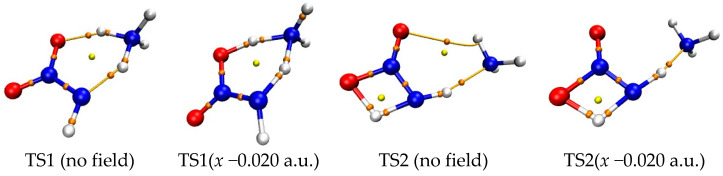
Selected bond critical point (BCP) of AIM in TS1 and TS2.

**Figure 3 molecules-28-02586-f003:**
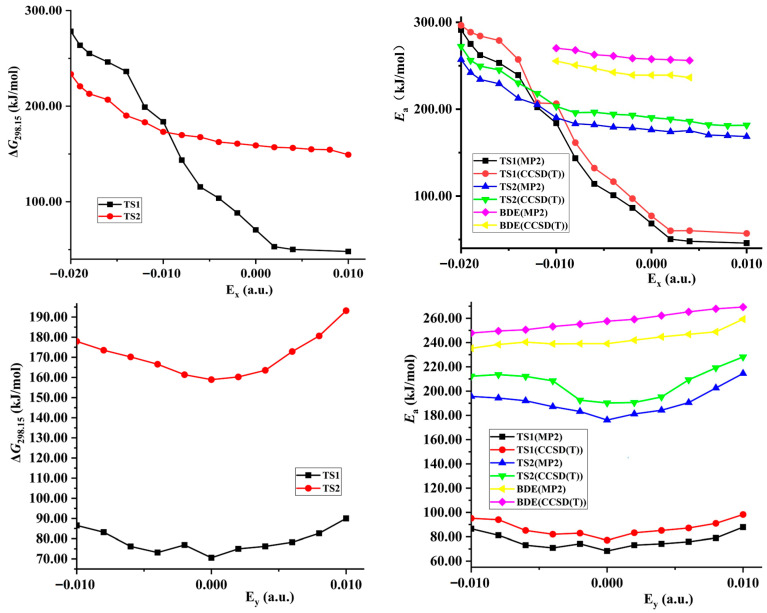
Gibbs energies (∆*G*), barriers (*E*_a_), or N−NO_2_ BDEs versus field strengths along the different field orientations (E_x_, E_y_ and E_z_).

**Figure 4 molecules-28-02586-f004:**
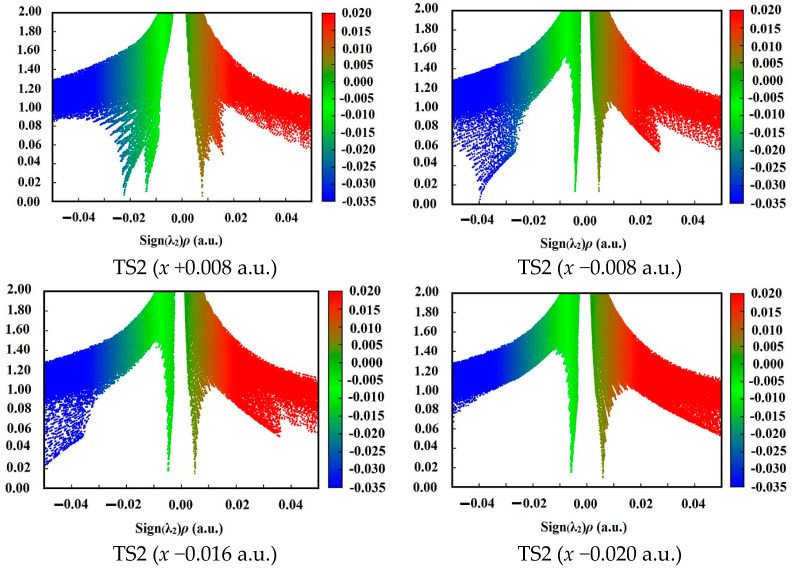
Plots of the RDG versus the electron density multiplied by the sign of the second Hessian eigenvalue (λ_2_) for TS1 and TS2. Green indicates strong attractive interaction, blue-brown indicates weak attractive interaction or vdW interaction, and red indicates steric effect.

**Figure 5 molecules-28-02586-f005:**
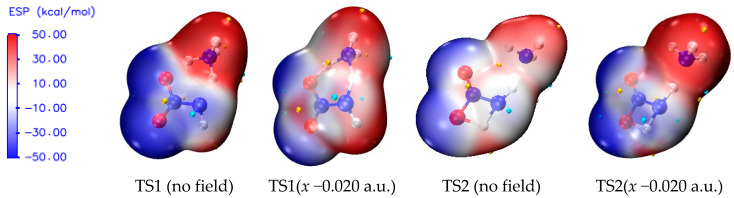
Surface electrostatic potentials of TS1 and TS2 in the different field strengths (a.u.) along −*x*-orientation. (Small gold and blue spheres represent the most positive and most negative surface electrostatic potentials, respectively.).

**Table 1 molecules-28-02586-t001:** Corrected reaction rate constants (*k*_298.15 K,C_ and *k*_688 K,C_, s^−1^) in the absence and presence of fields of varying strengths and directions for the intermolecular hydrogen exchange path (TS1) and 1,3-intramolecular hydrogen transference path (TS2) at the MP2/6-311++G(2d,p) level.

Field	*k*_298.15 K,C_ (TS1)	*k*_688 K,C_ (TS1)	*k*_298.15 K,C_ (TS2)	*k*_688 K,C_ (TS2)
No field	3.24 × 10^0^	2.74 × 10^6^	4.48 × 10^−15^	5.63 × 10^−1^
*z* −0.010	1.59 × 10^0^	1.55 × 10^6^	2.92 × 10^−26^	1.58 × 10^−6^
*z* −0.008	3.19 × 10^−1^	1.39 × 10^6^	1.92 × 10^−22^	3.99 × 10^−4^
*z* −0.006	9.49 × 10^0^	4.73 × 10^6^	2.98 × 10^−20^	2.78 × 10^−3^
*z* −0.004	2.20 × 10^0^	5.97 × 10^6^	1.67 × 10^−17^	2.16 × 10^−2^
*z* −0.002	1.78 × 10^0^	4.32 × 10^6^	2.64 × 10^−15^	3.65 × 10^−1^
*z* +0.002	1.20 × 10^0^	2.60 × 10^6^	8.69 × 10^−15^	9.27 × 10^−1^
*z* −0.004	1.90 × 10^0^	2.00 × 10^6^	4.52 × 10^−14^	1.47 × 10^0^
*z* −0.006	7.90 × 10^−1^	1.39 × 10^6^	5.19 × 10^−14^	2.65 × 10^0^
*z* −0.008	1.38 × 10^0^	3.39 × 10^6^	7.96 × 10^−13^	3.53 × 10^0^
*z* −0.010	3.09 × 10^−1^	8.13 × 10^5^	3.13 × 10^−12^	9.80 × 10^0^
*y* −0.010	5.10 × 10^−3^	6.10 × 10^4^	2.14 × 10^−18^	1.12 × 10^−2^
*y* −0.008	1.96 × 10^−2^	1.25 × 10^5^	1.27 × 10^−17^	3.79 × 10^−2^
*y* −0.006	3.33 × 10^−1^	8.95 × 10^5^	4.75 × 10^−17^	4.28 × 10^−2^
*y* −0.004	1.13 × 10^0^	2.49 × 10^6^	2.06 × 10^−16^	1.77 × 10^−1^
*y* −0.002	2.51 × 10^−1^	1.43 × 10^6^	1.67 × 10^−15^	2.91 × 10^−1^
*y* +0.002	5.33 × 10^−1^	1.22 × 10^6^	2.63 × 10^−15^	5.17 × 10^−1^
*y* +0.004	3.21 × 10^−1^	8.67 × 10^5^	6.98 × 10^−16^	1.60 × 10^−1^
*y* +0.006	1.42 × 10^−1^	1.80 × 10^6^	1.61 × 10^−17^	3.01 × 10^−2^
*y* +0.008	2.45 × 10^−2^	1.23 × 10^5^	7.19 × 10^−19^	6.55 × 10^−3^
*y* +0.010	1.25 × 10^−3^	8.81 × 10^4^	4.55 × 10^−21^	2.73 × 10^−4^
*x* +0.010	2.71 × 10^4^	2.76 × 10^8^	2.3 × 10^−13^	6.55 × 10^0^
*x* +0.008			2.8 × 10^−14^	1.65 × 10^0^
*x* +0.006			2.29 × 10^−14^	1.06 × 10^0^
*x* +0.004	1.15 × 10^4^	3.67 × 10^8^	1.26 × 10^−14^	7.85 × 10^−1^
*x* +0.002	3.77 × 10^3^	2.79 × 10^8^	9.51 × 10^−15^	6.83 × 10^−1^
*x* −0.002	2.61 × 10^−3^	4.45 × 10^4^	2.12 × 10^−15^	4.65 × 10^−1^
*x* −0.004	5.80 × 10^−6^	2.55 × 10^3^	1.04 × 10^−15^	2.31 × 10^−1^
*x* −0.006	5.58 × 10^−8^	4.73 × 10^2^	1.32 × 10^−16^	1.19 × 10^−1^
*x* −0.008	8.09 × 10^−13^	3.50 × 10^−1^	5.46 × 10^−17^	4.58 × 10^−2^
*x* −0.010	6.21 × 10^−20^	8.71 × 10^−5^	1.36 × 10^−17^	2.76 × 10^−2^
*x* −0.012	1.06 × 10^−22^	2.09 × 10^−6^	2.39 × 10^−19^	2.69 × 10^−3^
*x* −0.014	5.96 × 10^−29^	2.26 × 10^−9^	1.45 × 10^−20^	8.11 × 10^−4^
*x* −0.016	1.13 × 10^−30^	5.45 × 10^−13^	1.71 × 10^−23^	1.48 × 10^−5^
*x* −0.018	4.40 × 10^−32^	2.98 × 10^−13^	1.42 × 10^−24^	1.05 × 10^−5^
*x* −0.019	1.56 × 10^−33^	4.11 × 10^−15^	1.07 × 10^−27^	4.52 × 10^−6^
*x* −0.020	4.15 × 10^−36^	2.13 × 10^−15^	3.44 × 10^−28^	2.92 × 10^−6^

**Table 2 molecules-28-02586-t002:** The surface electrostatic potentials of the oxygen atoms ($V_{S,O8}^{-}$, kcal/mol), average positive and negative values of the surface potentials ($\bar{V_{S}^{+}}$, $\bar{V_{S}^{-}}$, kcal/mol), and their variances ($\sigma_{+}^{2}$ and $\sigma_{-}^{2}$ (kcal/mol)^2^), as well as polar surface area (PSA, %) for TS1 and transition state NH_2_NO_2_∙∙∙H_2_O→NHN(O)OH∙∙∙H_2_O (TS) in the different field strengths (a.u.) along −*x*-orientation at the MP2/6-311++G(2d,p) level.

Field	${V_{S,O8}^{-}}_{\mathrm{(TS1)}}$	${\bar{V_{S}^{+}}}_{\mathrm{(TS1)}}$	${\bar{V_{S}^{-}}}_{\mathrm{(TS1)}}$	${\sigma_{+}^{2}}_{\mathrm{(TS1)}}$	${\sigma_{-}^{2}}_{\mathrm{(TS1)}}$	PSA(%)	${{V_{S,O6}^{-}}_{\mathrm{(TS)}\;}}^{a}$	${\bar{V_{S}^{+}}}_{\mathrm{(TS)}}$	${{\bar{V_{S}^{-}}}_{\mathrm{(TS)}\;}}^{a}$	${{\sigma_{+}^{2}}_{\mathrm{(TS)}}}^{\;a}$	${{\sigma_{-}^{2}}_{\mathrm{(TS)}}}^{\;a}$	PSA_(TS)_
No Field	−55.8	34.8	−28.2	381.9	264.3	82.7	−23.2	22.0	−16.1	235.9	93.5	73.2
*x* −0.002	−54.4	33.1	−27.5	346.8	254.8	81.7	−22.4	21.5	−17.3	231.8	89.5	72.1
*x* −0.004	−53.0	31.5	−26.6	310.2	246.0	80.6	−20.3	20.8	−16.5	220.3	78.3	70.8
*x* −0.006	−51.7	29.8	−25.8	278.7	223.5	78.3	−21.6	19.6	−16.6	218.1	65.7	70.3
*x* −0.008	−50.7	32.5	−27.2	251.6	190.7	83.9	−20.3	21.3	−15.1	206.2	58.3	69.5
*x* −0.010	−49.6	34.4	−28.2	279.9	146.2	90.6	−19.7	19.7	−12.2	223.4	51.4	68.8
*x* −0.012	−47.8	35.2	−26.7	288.3	122.8	90.8	−30.2	22.5	−13.6	218.7	72.1	68.1
*x* −0.013							−29.8	19.3	−12.8	201.3	63.6	67.3
*x* −0.014	−44.9	32.6	−24.3	250.2	109.7	88.4	−29.5	20.2	−15.1	212.9	55.0	67.2
*x* −0.015							−28.2	21.3	−12.7	197.9	49.8	65.2
*x* −0.016	−41.2	28.7	−21.6	211.8	111.3	84.1						
*x* −0.018	−37.6	22.0	−19.9	167.6	102.5	76.5						
*x* −0.019	−35.0	19.3	−18.2	139.2	91.3	70.8						
*x* −0.020	−33.1	16.7	−15.6	116.9	90.1	68.0						

^a^ From Ref. [30].

## Data Availability

The data related to this research can be accessed upon reasonable request via email.

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
