# Peer review of "Strong External Electric Fields Reduce Explosive Sensitivity: A Theoretical Investigation into the Reaction Selectivity in NH2NO2∙∙∙NH3"

_molecules, 2023, doi:10.3390/molecules28062586_

Round 1

Reviewer 1 Report

This manuscript deals with the effects of external electric fields on the reactions in NH2NO2∙∙∙NH3 model system of the nitroamine explosive with alkaline additive, investigated using electronic structure calculations. The rate constants are calculated with conventional transition state theory. Tunneling effects are considered using Wigner method. It is well performed and the data in the manuscript supports conclusions. However, it requires some improvements. This paper is publishable if authors address the comments in the revised version.

Comments:

1. Page 1, line 16: “CCSD/6-311++G(2d,p) level” should be changed to “CCSD(T)/6-311++G(2d,p)”

2. It is helpful if the authors provide some justifications for selecting the MP2/6-311++G(2d,p) level for reactant and transition state optimization in the absence and presence of the different external electric fields. The reason for choosing the method MP2 should be given in the manuscript and cite some references. The author should also provide some reasons or supports for the choice of CCSD(T) method (For example: J. Chem. Theory Comput. 2013, 9, 4403−4413.; J. Phys. Chem. A 2021, 125, 5963−5975.)

3. Page 3, line 136-137: Please give some indication of the expected accuracy of the calculations for the level of theory used in this study

4. Page 4, line 170-171: It would be helpful to the readers if authors provide structure of another conformation of NH2NO2∙∙∙NH3 and its binding energy in the supporting information.

5. My major concern pertains to the potential impact of spin contamination and multireference character on the reactant, transition states in the different external electric field strengths and orientations (including those in the absence of field) at the MP2/6-311++G(2d,p) and CCSD(T)/6-311++G(2d,p)//MP2/6-311++G(2d,p) level. The reactant and TSs obtained in this work significantly influenced by these limitations. I suggest including a discussion on this matter in the manuscript.

6. Page 5, section 3.2 (and elsewhere): What are the estimated uncertainties in the calculated kinetics?

7. Please give the Cartesian coordinates for the optimized geometries of reactant and transition states in the different external field strengths and orientations as well as in the absence of field in the supplementary material.

Author Response

A Response to the Referee Report 1

Thank Editor very much for giving us a chance to revise our manuscript! Thank gratefully Reviewer for providing many helpful suggestions to our manuscript (ID: molecules-2247143)!

    Thank you very much for your seriousness, responsibility and professionalism! In particular for the suggestions on the “selection of calculation method”, “accuracy of the calculations for the level of theory” and “the potential impact of spin contamination and multireference character on the reactant, transition states in the different external electric field”. These will be of great help to our manuscript and future work.

According to reviewer, this manuscript deals with the effects of external electric fields on the reactions in NH2NO2∙∙∙NH3 model system of the nitroamine explosive with alkaline additive, investigated using electronic structure calculations. The rate constants are calculated with conventional transition state theory. Tunneling effects are considered using Wigner method. It is well performed and the data in the manuscript supports conclusions. However, it requires some improvements. This paper is publishable if authors address the comments in the revised version.

Answer: Thank you very much!

Comments:

  1. Page 1, line 16: “CCSD/6-311++G(2d,p) level” should be changed to “CCSD(T)/6-311++G(2d,p)”

Answer: “CCSD/6-311++G(2d,p) level” has been corrected to “CCSD(T)/6-311++G(2d,p)”.

  1. It is helpful if the authors provide some justifications for selecting the MP2/6-311++G(2d,p) level for reactant and transition state optimization in the absence and presence of the different external electric fields. The reason for choosing the method MP2 should be given in the manuscript and cite some references. The author should also provide some reasons or supports for the choice of CCSD(T) method (For example: J. Chem. Theory Comput. 2013, 9, 4403−4413.; J. Phys. Chem. A 2021, 125, 5963−5975.)

Answer: We thank gratefully Reviewer for this suggestion. According to reviewer, we revise it as follows:

Recently, we have used the second order Møller–Plesset perturbation theory (MP2) [45] and coupled cluster theory with single, double and perturbative triple substitutions CCSD(T) [46] methods with the 6-311++G(2d,p) basis set to investigate the effects of the external electric fields on the hydrogen transference kinetics of the model molecules of energetic material [30,35,36], and found that the bond dissociation energies (BDEs) of the “trigger linkages” [7] and barrier heights of the initiation reactions associated with hydrogen atom transfer as well as explosive sensitivities [47,48] could be accurately calculated and estimated in the absence and presence of the different external electric fields. Two methods were also used to quantify the electric field effect on the intermolecular interactions [49,50]. Therefore, in this work, the molecular geometries of the reactant and transition state (TS) were fully optimized using the MP2/6-311++G(2d,p) method in the absence and presence of the external electric fields. The energy minima were judged by the criteria of lacking imaginary frequency or only one imaginary frequency in which two atoms vibrate along the direction of forming or breaking chemical bonds. The activation energies (Ea) were calculated at the MP2/6-311++G(2d,p) and CCSD(T)/6-311++G(2d,p)//MP2/6-311++G(2d,p) level, respectively. See “the 3rd paragraph in Page 3” in “2. Computational details”.

  1. Page 3, line 136-137: Please give some indication of the expected accuracy of the calculations for the level of theory used in this study.

Answer: We thank gratefully Reviewer for this suggestion. This is very helpful for our future work! According to reviewer, we revise it as follows:

CCSD(T) is a high level electron correlation method [51−53], within the chemical accuracy of 1.0 kcal/mol in the total energies [54]. It can be thought of as the “gold standard” in computational chemistry for single-reference systems, in particular in the kinetics [55]. Nowadays, the CCSD(T) method has served as a reference in almost any computational approach if affordable, combined with sufficiently large and diffuse basis sets. For example for the hydrogen bonded systems, the accuracy of CCSD(T) is rather high in the dissociation energy and the error is much lower than for atomization energies, with a RMS error of 0.17 kJ/mol and a relative RMS error of 0.6%, which is a composite method akin to W4 theory. MP2 yields RMS errors of around 1.3 kJ/mol and 6.5%, almost 10 times those of CCSD(T) [56]. Since the external electric field acts on the dipole moment, for MP2 and CCSD(T), the influence of the external electric field on the accuracy is equivalent to the first-order perturbation of energy. See “the last paragraph in Page 3 and the 1st paragraph in Page 3” in “2. Computational details”.

Moreover, the spin-network-scaled MP2 (SNS-MP2) method has been trained on a new data set consisting of over 200 000 complete basis set (CBS)-extrapolated coupled-cluster interaction energies, which are considered the gold standard for chemical accuracy. SNS-MP2 predicts gold-standard binding energies of unseen test compounds with a mean absolute error of 0.04 kcal mol-1 (root-mean-square error 0.09 kcal mol-1), a 6- to 7-fold improvement over MP2 [The Journal of Chemical Physics 147, 161725 (2017); doi: 10.1063/1.4986081]. Since method SNS-MP2 is not used in our manuscript, it is not mentioned in our manuscript.

  1. Page 4, line 170-171: It would be helpful to the readers if authors provide structure of another conformation of NH2NO2∙∙∙NH3 and its binding energy in the supporting information.

Answer: We thank gratefully Reviewer for this suggestion. The structure of another conformation of NH2NO2∙∙∙NH3 and its binding energy have been provided in the supporting information. See Figure S2.

  1. My major concern pertains to the potential impact of spin contamination and multireference character on the reactant, transition states in the different external electric field strengths and orientations (including those in the absence of field) at the MP2/6-311++G(2d,p) and CCSD(T)/6-311++G(2d,p)//MP2/6-311++G(2d,p) level. The reactant and TSs obtained in this work significantly influenced by these limitations. I suggest including a discussion on this matter in the manuscript.

Answer: We thank gratefully Reviewer for this suggestion. This is very helpful for our future work! According to reviewer, we revise it as follows:

It is well known that the unrestricted single reference methods give rise to spin-contamination when applied to open-shell systems since the unrestricted Hartree-Fock wave function is not an eigenfunction of the total spin operator, i.e., the expectation value <S2> may not be equal to S(S+1), leading to an inaccurate energy. For the restricted open-shell Hartree-Fock calculations with the right <S2>, the unphysical results are often generated due to the artificially ruined spin polarization. Therefore, it is necessary to use multireference methods for highly spin-contaminated systems with the inherently multireference nature, especially for the transition state [68,69]. For the open-shell systems calculated by the MP2 and CCSD methods, although the treatment of the electron correlation should lower spin contamination, the spin contamination also occurs at the correlated level [70]. Furthermore, although the CCSD method significantly outperform MP2 in describing systems with a strongly spin-contaminated reference because the cluster single excitation operator partly accounts for the orbital relaxation effects [71−73], CCSD(T) are often rather sensitive to the spin contamination of the reference [74,75]. In this work, for the single point energy calculation on the reactants and transition states for the hydrogen exchanges in TS1 and TS2 at the UMP2/6-311++G(2d,p) and UCCSD(T)/6-311++G(2d,p)//MP2/6-311++G(2d,p) level, we obtained <S2> ≈ 0.0 (vs expected S(S+1)=0) in the different external electric field strengths and orientations, including those in the absence of field. Therefore, the impact of the spin contamination and multireference character on the initiation reactions in NH2NO2∙∙∙NH3 in the external electric field as well as in the absence of field can be ignored. See “the 1st paragraph in Page 8”.

  1. Page 5, section 3.2 (and elsewhere): What are the estimated uncertainties in the calculated kinetics?

Answer: We thank gratefully Reviewer for this suggestion. According to reviewer, we revise it as follows:

Indeed, the use of the word “uncertainty” obscures the meaning of the sentence. From the context, this sentence means that the change of activation distance implies that the reaction path is not unique under the “strong” external electric fields. Therefore, according to the reviewer's suggestion, the word “uncertainty” has been deleted and “…lead to uncertainty or diversification of the…”has been corrected to “…lead to uncertainty or diversification of the…”. The sentence “ i.e., either H5 or H7 transfer plays a key role in the main reaction” has been added.

  1. Please give the Cartesian coordinates for the optimized geometries of reactant and transition states in the different external field strengths and orientations as well as in the absence of field in the supplementary material.

Answer: We thank gratefully Reviewer for this suggestion. According to reviewer, the Cartesian coordinates for the optimized geometries of reactant and transition states in the different external field strengths and orientations as well as in the absence of field have been given in the supplementary material. Unfortunately, several “out. Files” were flushed and lost because about 20 people were using the computing workstation at the same time. But the important “out. Files” and calculation results are not lost, and they were added in the supplementary material.

In addition, with the help of a native English expert, the grammar has been comprehensively revised.

All the “References” in the reply letter are placed in our manuscript, which is completely consistent with that in our manuscript.

30 references have been added, and they are in RED.

We hope gratefully that more helpful suggestions and comments will be provided! Thank you! 

Sincerely,

Fu-de Ren

                                      March 4, 2023  

Reviewer 2 Report

The present manuscript by Fu De Ren et al. deals with high quality calculations for the reaction mechanisms regarding the dissociation of nitrogen containing compounds under various electric fields and their influence in the explosiveness of this compound mixture as correlated to various physicochemical values associated with the reaction kinetics and thermodynamics. 

The presented calculations are made with rigor and at high levels of theory. From a quick search of the corresponding authors' work it would seem that the topic under study is a logical step in the direction of their research lines, and their conclusions--albeit written in a bit convoluted fashion--are solid and strong. However, there are several problems with the manuscript that make it almost unreadable, confusing and tiresome, and which should be corrected before considering its publication. 

To begin with, the document was extremely hard to follow until I printed figure 1 and kept it next to me for the whole duration of my reading. More figures representing each of the mechanisms (both with the structures already provided and corresponding energy profiles) at appropriate stages would greatly benefit the reading experience. Figure 1 is too crammed with information. I would suggest to use a separate figure just to illustrate the coordinate system alone. 

Also, tables provided in the body of the manuscript would be better allocated in the SI while a corresponding graph or plot could serve the readers better for visualizing trends. 

Pages 2-3, lines 98-102 read: "They found that the decomposition rate of RDX was hardly affected in the neutral medium, [...] while it was increased in the alkaline medium, [...]. We have found that, in alkaline environment, the explosive stability was changed greatly [...]” This sentences are confusing. What do the authors mean by 'changed greatly'? Was it greatly decreased or increased greatly? Reports should be specific and not to be left out to interpretation, specially to avoid confusion with the first sentence. 

Page 3 lines 113-115 The sentence would appear incomplete; I would advise to redraft it. 

Computational details. It has become a good practice to provide the full route section (when working with Gaussian) for others to fully reproduce results, whether as a section in the SI or as part of this section. The inclusion of an electric field is not necessarily a standard procedure in computational chemistry, therefore novice comp chemists would greatly benefit from reading how to do it. 

Page 3 lines 140-141. There is an error in the axis labeling. "... and thus the y- axis is defined [...] perpendicular to the xy plane". In this regard, the use of a hyphen for the axis labeling is utterly confusing since it sometimes is typed as -y and others as y-, and even worse when we read +x-.

Page 4 line 156 Why is the value T=688 K (414.85 °C) used? what is so special about this particular temperature? Please specify. 

Page 4 lines 162 and 163 There is an inconsistency with the symbols used for frequency; it would appear the text uses the latin letter v, whereas the (unnumbered) equation uses the greek letter nu. Please homogenize. 

Page 4 line 172. The 32 kJ/mol difference in stability could be misleading unless its clarified: Is this the electronic energy difference? then stability should not be invoked, unless this is the difference in the enthalpy of formation. Please specify. 

Page 5 lines 178-179 (and page 9 lines 300-301) The tautomeric equilibria would be clearer as separate equations in separate lines. Please correct. 

Page 5 line 303 reads: "the average amplitude of H in the imaginary frequency", for the sake of correctness the word amplitude should be reserved for vibrations, instead I ask the authors change the word amplitude for 'displacement' in this and all other instances. 

Page 10 line 330. The Mulliken population analysis and charges should be deprecated by now. I strongly suggest the authors to report a better charge treatment. 

Conclusions should be summarized. What are the directions of the field that allow for the stabilization of the explosive mixtures? For which mechanisms? Now, the biggest question I get from this reading is how can we account for the random orientations these molecules adopt in the exploding device? since they're not crystalline solids, there seems to be no way to align an electric field effectively to keep the mixture stable but rather only a portion of the randomly aligned molecules, the rest of which could have their stability reduced and have the device explode. 

I would advise against the use of vernaculars such as 'holy grail' or 'idiot's daydream' not just because I strongly believe scientific reporting should be a serious endeavor which can be skewed and biased by authors' opinions embedded as subjective adjectives, but also because the more neutral the language the more accessible the article is for those who are not experts, or not even proficient for that matter, in English. 

Other mistakes include "strengthening the strength of the field", (why not simply saying "increasing the field strength" or something along those lines?) among others. 

Author Response

A Response to the Referee Report 2

Thank Editor very much for giving us a chance to revise our manuscript! Thank gratefully Reviewer for providing many helpful suggestions to our manuscript (ID: molecules-2247143)!

Thank you very much for your seriousness, responsibility and professionalism! I can feel that even for every word and punctuation, you are very careful to help us check and verify! I sincerely thank you! This will be of great help to our manuscript and future work.

According to reviewer, “The present manuscript by Fu De Ren et al. deals with high quality calculations for the reaction mechanisms regarding the dissociation of nitrogen containing compounds under various electric fields and their influence in the explosiveness of this compound mixture as correlated to various physicochemical values associated with the reaction kinetics and thermodynamics”.

Answer: Thank you very much!

According to reviewer, “The presented calculations are made with rigor and at high levels of theory. From a quick search of the corresponding authors' work it would seem that the topic under study is a logical step in the direction of their research lines, and their conclusions--albeit written in a bit convoluted fashion--are solid and strong. However, there are several problems with the manuscript that make it almost unreadable, confusing and tiresome, and which should be corrected before considering its publication”.

Answer: We thank gratefully Reviewer for this suggestion “…conclusions--albeit written in a bit convoluted fashion almost… unreadable, confusing and tiresome”. This is very helpful for our future work! According to reviewer, we revise it as follows:

Firstly, with the help of a native English expert, the grammar has been comprehensively revised. All the revision are in RED.

Secondly, it is the most important to revise the conclusion on “albeit written in a bit convoluted fashion almost”. In order to draw a reasonable conclusion, according to the reviewer, we revise it as follows:

We added a discussion: Noted that, the imposed external field strengths cannot exceed the dielectric strengths of the explosive system. Otherwise, the explosive device may be broken down, and the work to control the selectivity of the reaction and reduce the explosive sensitivity by adjusting the external electric fields will not be practical. Furthermore, although theoretically the sensitivity can be reduced by adjusting the external field orientation and strengths, in practice, the technology may be very difficult because the direction of molecular motion is disorderly, and the control of the direction of the external electric field to the “reaction axis” is one of the technical bottlenecks. There seems to be no way to align an electric field effectively to keep the mixture stable but rather only a portion of the randomly aligned molecules, the rest of which could have their stability reduced and have the device explode. However, the external electric field effect itself of changing the reaction pathways and explosive sensitivity should be remarkable enough to be submitted as the subject of experimental scrutiny in the energetic material field. After all, the orientation of molecules can be achieved by means of Langmuir-Blodgett techniques [79, 80], and there has been the experimental work in which the external electric field is imposed to energetic material [81]. Furthermore, the external electric field can drive the proton transfer [82], change the reaction pathways between different tautomers [83] and control the selectivity of bond activations [84]. Electron transport studies in the molecular-scale systems have become possible [85, 86], not only in theory [87] but also in experiment [88]. See “the 2nd paragraph in Page 14”.

Thus, we added a paragraph in “conclusion”: It should be noted that, although theoretically the sensitivity can be reduced by adjusting the external field orientation and strengths, in practice the technology is very difficult because the direction of molecular motion is disorderly, and the control of the direction of the external electric field to the “reaction axis” is one of the technical bottlenecks. There seems to be no way to align an electric field effectively to keep the mixture stable but rather only a portion of the randomly aligned molecules, the rest of which could have their stability reduced and have the device explode. However, the external electric field effect itself of changing the reaction pathways and explosive sensitivity should be remarkable enough to be submitted as the subject in the energetic material field. After all, the orientation of molecules can be achieved, and the external electric field can drive the proton transfer and control the selectivity of bond activations. See “the 3rd paragraph in Page 17”.

(1) According to reviewer, “To begin with, the document was extremely hard to follow until I printed figure 1 and kept it next to me for the whole duration of my reading. More figures representing each of the mechanisms (both with the structures already provided and corresponding energy profiles) at appropriate stages would greatly benefit the reading experience. Figure 1 is too crammed with information. I would suggest to use a separate figure just to illustrate the coordinate system alone”.

Answer: We thank gratefully Reviewer for this suggestion! According to reviewer, a separate figure has been used to illustrate the coordinate system (see the 4th figure in Fig. 1). 

(2) According to reviewer, “Also, tables provided in the body of the manuscript would be better allocated in the SI while a corresponding graph or plot could serve the readers better for visualizing trends”.

Answer: Thank you very much for your this suggestions! According to the reviewer, we revise it as follows:

In the original Tables 1 and 2, the most important information was “Gibbs energies (∆G), barriers (Ea) and N−NO2 BDEs under the different field strengths along the different field orientations (Ex, Ey and Ez)”, while the other information, such as the imaginary frequencies, reaction rate constants and Wigner tunneling corrections (k),was collected in the Tables 1 and 2 to give the way and result of calculating the corrected reaction rate constants. These important information are all shown in Figure 2. Therefore, according to the reviewer, in order to keep the manuscript concise, Table 1 and Table 2 were moved to the annex.

Because the differences between the values of the corrected reaction rate constants in the different paths or temperatures or field orientations are dozens of orders of magnitude, while it very small in some in a certain interval. For example, for k298.15 K,C (TS1), the values is from 1.59×100 to 4.15×10-36. It is very difficult to show the trend and values of the corrected reaction rate constants vs. effects of the external fields by the figures. For example, most of them are shown by the “0.0” value. Many efforts have failed in the end. Therefore, the corrected reaction rate constants for TS1 and TS2 have to be collected in the Table 1. Table 2 was combined with Table 1. 

See Tables 1, S3 and S6.

(3) According to reviewer, “Pages 2-3, lines 98-102 read: "They found that the decomposition rate of RDX was hardly affected in the neutral medium, [...] while it was increased in the alkaline medium, [...]. We have found that, in alkaline environment, the explosive stability was changed greatly [...]” This sentences are confusing. What do the authors mean by 'changed greatly'? Was it greatly decreased or increased greatly? Reports should be specific and not to be left out to interpretation, specially to avoid confusion with the first sentence”.

Answer: Thank you very much for your this suggestions! According to the reviewer, we revise it as follows:

They found that the decomposition rate of RDX was hardly affected in the neutral medium, such as benzene, isooctane and naphthalene, etc., and compared with molten RDX, the change of rate is less than 5.0% [37]. While it was increased by 2.8~21.0 times in the alkaline medium, such as the agent with the ‒NH2 or ‒NHR group [39]. We found that, in alkaline environment, the impact sensitivities of explosives were even increased or decreased by more than 20% due to either the modified BDEs of the “trigger bonds”…….

See “the last paragraph in Page 2 and the 1st paragraph in Page 3”.

(4) According to reviewer, “Page 3 lines 113-115 The sentence would appear incomplete; I would advise to redraft it”.

Answer: Thank you very much for your this suggestions! According to the reviewer, we revise it as follows:

The fragments of the initial decomposition reaction of NH2NO2 were determined as O•, NH2NO•, NH2N•, •NH2, •NO2, NO2 and NO [42], and the competitive reaction between the N‒NO2 bond cleavage and NH2NO2 → NH2ONO rearrangement was also confirmed [43]. See “the 2nd paragraph in Page 3”.

(5) According to reviewer, “Computational details. It has become a good practice to provide the full route section (when working with Gaussian) for others to fully reproduce results, whether as a section in the SI or as part of this section. The inclusion of an electric field is not necessarily a standard procedure in computational chemistry, therefore novice comp chemists would greatly benefit from reading how to do it”.

Answer: Thank you very much for your this suggestions! According to the reviewer, we revise it as follows:

The calculation process is as follows:

Figure S1   The research scheme for the effects of external electric fields on the initiation reactions in NH2NO2∙∙∙NH3

(6) According to reviewer, “Page 3 lines 140-141. There is an error in the axis labeling. "... and thus the y- axis is defined [...] perpendicular to the xy plane". In this regard, the use of a hyphen for the axis labeling is utterly confusing since it sometimes is typed as -y and others as y-, and even worse when we read +x-”.

Answer: Thank you very much for your this suggestions! According to the reviewer, we revise it as follows:

“-y” means the negative direction of the y-axis, the same for others, while “y-” means the hyphen for the axis labeling. In order to distinguish negative direction from the hyphen for the axis labeling, the sign of “negative sign” is elongated throughout the manuscript.

According to Reviewer, the x-axis is defined as that the x-axis is in the plane composed of the N atom in the −NH2 group, and two O atoms in the −NO2 group, and the N atom of NH3 is approximately along the −x-axis. The sentence “y-axis is defined as the direction perpendicular to the xy-plane” has been corrected to “y-axis is defined as the direction perpendicular to the xz-plane”.

See “the 2nd paragraph in Page 4”.

(7) According to reviewer, “Page 4 line 156 Why is the value T=688 K (414.85 °C) used? what is so special about this particular temperature? Please specify”.

    Answer: Thank you very much for your this suggestions! According to the reviewer, we revise it as follows:

Strictly speaking, there is nothing special about this temperature. For the experimental determination of the detonation temperature of nitramine explosives, generally starting from the temperature of 298 K, the data is recorded every 5 K, and when the data recorded 130 times, the temperature is just 688 K. In our previous papers, we have calculated the reaction rate at this maximum temperature (Fu-de Ren, Duan-lin Cao, Wen-jing Shi, Min You, A dynamic prediction of stability for nitromethane in external electric field, RSC Adv., 2017, 7, 47063–47072; Fu-de Ren, Duan-lin Cao, Wen-jing Shi, A dynamics prediction of nitromethane → methyl nitrite isomerization in external electric field, J Mol Model (2016) 22: 96; Fu-de Ren, Wen-jing Shi, Duan-lin Cao, Yong-xiang Li, De-hua Zhang, Xian-feng Wang, Zhao-yang Shi, External electric field reduces the explosive sensitivity: a theoretical investigation into the hydrogen transference kinetics of the NH2NO2∙∙∙H2O complex, J Mol Model (2020) 26: 351). Therefore, in this manuscript, the value T=688 K (414.85 °C) was used.

(8) According to reviewer, “Page 4 lines 162 and 163 There is an inconsistency with the symbols used for frequency; it would appear the text uses the latin letter v, whereas the (unnumbered) equation uses the greek letter nu. Please homogenize”.

Answer: Thank you very much for your this suggestions! According to the reviewer, “v” has been corrected to “v”.

(9) According to reviewer, “Page 4 line 172. The 32 kJ/mol difference in stability could be misleading unless its clarified: Is this the electronic energy difference? then stability should not be invoked, unless this is the difference in the enthalpy of formation. Please specify”.

Answer: Thank you very much for your this suggestions! According to the reviewer, we revise it as follows:

From the “.out File”, EUMP2=-316.9937146 a.u. for the former, and EUMP2=-316.9815353 a.u. for the latter.Therefore,  ΔE(Tot., EUMP2)=31.98 kJ/mol.

According to Reviewer, this sentence has been corrected to “The electron energy of the former is 32.0 kJ/mol less than that of the latter at the MP2/6-311++G(2d,p) level”. See “the 3rd paragraph in Page 5”.

(10) According to reviewer, “Page 5 lines 178-179 (and page 9 lines 300-301) The tautomeric equilibria would be clearer as separate equations in separate lines. Please correct”.

    Answer: Thank you very much for your this suggestions! According to Reviewer, the tautomeric equilibria have been corrected as separate equations in separate lines. See “the 5th paragraph in Page 5”.

(11) According to reviewer, “Page 5 line 303 reads: "the average amplitude of H in the imaginary frequency", for the sake of correctness the word amplitude should be reserved for vibrations, instead I ask the authors change the word amplitude for 'displacement' in this and all other instances”.

Answer: Thank you very much for your this suggestions! According to Reviewer, the word “amplitude” has been corrected to “displacement”.

(12) According to reviewer, “Page 10 line 330. The Mulliken population analysis and charges should be deprecated by now. I strongly suggest the authors to report a better charge treatment.

Answer: Thank you very much for your this suggestions! According to Reviewer, The Mulliken population analysis has been deleted, and the APT analysis has been used to reveal the essence of the concerted reaction between the N6→H7→O8 and N1→H5→N6 reactions, as well as 1,3-intramolecular hydrogen transfer in external electric field.

Under the external electric field, the change of the atomic charge is often complicated due to the influence of the molecular dipole (internal electric field). From the APT charges collected in Table S4, along the −x-direction with the field strengths no more than −0.008 a.u., both of the negative charges of N6 and O8 increase with the increase of the field strengths. When the field strengths are more than −0.008 a.u., the negative charges of O8 decrease while those of N6 decreases first (−0.008 a.u. ~ −0.012 a.u.) and then increases, leading the more notable negative charge of N6 than that of O8 in the field strength of −0.020 a.u.. Thus, the H7 atom with the positive charge would rather bind to N6 than O8 with the increase of the external electric field strength in the −x-direction. In other words, the −x-direction of the external electric field is opposite to that of the “reaction axis” along N6→H7→O8, and it is not beneficial to the hydrogen transfer from NH2NO2∙∙∙NH3 to NH2N(O)OH•âˆ™âˆ™âˆ™•NH2 with the increase of the electric field strength. Except for the field strengths of −0.008 a.u. ~ −0.012 a.u., the negative charges of N1 are always larger than those of N6, and thus the H7 atom with the positive charge would rather bind to N1 than N6. Therefore, although the −x-direction is the same as that of the N1→H5→N6 reaction axis, it is not beneficial to the intermolecular hydrogen transfer of NH2N(O)OH•âˆ™âˆ™âˆ™•NH2 → NHN(O)OH∙∙∙NH3, i.e., this electric field is unfavorable to the NH2NO2∙∙∙NH3→NHN(O)OH∙∙∙NH3 concerted reaction. It should be noted that, as mentioned above, in the absence of electric field, there is a concerted reaction between the N6→H7→O8 and N1→H5→N6 reactions, which promotes the intermolecular hydrogen exchange reaction of NH2NO2∙∙∙NH3→NHN(O)OH∙∙∙NH3. Thus, under the electric field along the −x-direction, the concerted effect is weakened, leading to the increased barrier heights of the hydrogen exchange reaction. Furthermore, the stronger the external electric field along the −x-direction, the more seriously the concerted effect is weakened and the higher the barrier heights become, as is in agreement with the barrier height results shown in Figure 3.

See “the last paragraph in Page 10 and the 1st paragraph in Page 11”.

(13) According to reviewer, “Conclusions should be summarized. What are the directions of the field that allow for the stabilization of the explosive mixtures? For which mechanisms? Now, the biggest question I get from this reading is how can we account for the random orientations these molecules adopt in the exploding device? since they're not crystalline solids, there seems to be no way to align an electric field effectively to keep the mixture stable but rather only a portion of the randomly aligned molecules, the rest of which could have their stability reduced and have the device explode”.

Answer: We thank gratefully Reviewer for this suggestion. This is very helpful for our future work! According to reviewer, we revise it as follows:

Judging from the initiation reaction of explosive at the molecular level, the sensitivity mainly depends on the barrier of the intermolecular hydrogen exchange. See “the last paragraph in Page 13”.

We added a discussion: Noted that, the imposed external field strengths cannot exceed the dielectric strengths of the explosive system. Otherwise, the explosive device may be broken down, and the work to control the selectivity of the reaction and reduce the explosive sensitivity by adjusting the external electric fields will not be practical. Furthermore, although theoretically the sensitivity can be reduced by adjusting the external field orientation and strengths, in practice, the technology may be very difficult because the direction of molecular motion is disorderly, and the control of the direction of the external electric field to the “reaction axis” is one of the technical bottlenecks. There seems to be no way to align an electric field effectively to keep the mixture stable but rather only a portion of the randomly aligned molecules, the rest of which could have their stability reduced and have the device explode. However, the external electric field effect itself of changing the reaction pathways and explosive sensitivity should be remarkable enough to be submitted as the subject of experimental scrutiny in the energetic material field. After all, the orientation of molecules can be achieved by means of Langmuir-Blodgett techniques [79, 80], and there has been the experimental work in which the external electric field is imposed to energetic material [81]. Furthermore, the external electric field can drive the proton transfer [82], change the reaction pathways between different tautomers [83] and control the selectivity of bond activations [84]. Electron transport studies in the molecular-scale systems have become possible [85, 86], not only in theory [87] but also in experiment [88]. See “the 2nd paragraph in Page 14”.

Thus, we added a paragraph in “conclusion”: It should be noted that, although theoretically the sensitivity can be reduced by adjusting the external field orientation and strengths, in practice the technology is very difficult because the direction of molecular motion is disorderly, and the control of the direction of the external electric field to the “reaction axis” is one of the technical bottlenecks. There seems to be no way to align an electric field effectively to keep the mixture stable but rather only a portion of the randomly aligned molecules, the rest of which could have their stability reduced and have the device explode. However, the external electric field effect itself of changing the reaction pathways and explosive sensitivity should be remarkable enough to be submitted as the subject in the energetic material field. After all, the orientation of molecules can be achieved, and the external electric field can drive the proton transfer and control the selectivity of bond activations. See “the 3rd paragraph in Page 17”.

(14) According to reviewer, “I would advise against the use of vernaculars such as 'holy grail' or 'idiot's daydream' not just because I strongly believe scientific reporting should be a serious endeavor which can be skewed and biased by authors' opinions embedded as subjective adjectives, but also because the more neutral the language the more accessible the article is for those who are not experts, or not even proficient for that matter, in English”.

Answer: We thank gratefully Reviewer for this suggestion. This is very helpful for our future work! According to reviewer, we revise it as follows:

According to Reviewer, the vernaculars “holy grail” and “idiot's daydream” have been deleted. With the help of a native English expert, the grammar has been comprehensively revised. All the revision are in RED.

(15) According to reviewer, “Other mistakes include "strengthening the strength of the field", (why not simply saying "increasing the field strength" or something along those lines?) among others”.

Answer: We thank gratefully Reviewer for this suggestion. According to reviewer, “strengthening the strength of the field” has been corrected to “increasing the field strength”. 

All the “References” in the reply letter are placed in our manuscript, which is completely consistent with that in our manuscript.

30 references have been added, and they are in RED.

We hope gratefully that more helpful suggestions and comments will be provided! Thank you! 

Sincerely,

Fu-de Ren

                                      March 4, 2023  

Round 2

Reviewer 1 Report

The authors have addressed my suggestions. I recommend publication in the present form.

Reviewer 2 Report

This new revised version of the manuscript by Fu De-Ren et al. is vastly superior to the previous one. Since authors have diligently answered all my concerns (and even more!) I consider it to be ready for publication in Molecules. Just one thing, when I asked for the route section (which is now provided as a scheme in the SI section) I meant the actual keywords used in the program, so other Gaussian users could replicate the methodology. Still, even if this change isn't made I think the manuscript is in top shape.